# Advances of Epigenetic Biomarkers and Epigenome Editing for Early Diagnosis in Breast Cancer

**DOI:** 10.3390/ijms23179521

**Published:** 2022-08-23

**Authors:** Pourya Sarvari, Pouya Sarvari, Ivonne Ramírez-Díaz, Frouzandeh Mahjoubi, Karla Rubio

**Affiliations:** 1Department of Clinical Genetics, National Institute of Genetic Engineering and Biotechnology, Tehran P.O. Box 14965/161, Iran; 2International Laboratory EPIGEN, Consejo de Ciencia y Tecnología del Estado de Puebla (CONCYTEP), Puebla 72160, Mexico; 3Facultad de Biotecnología, Campus Puebla, Universidad Popular Autónoma del Estado de Puebla (UPAEP), Puebla 72410, Mexico; 4Licenciatura en Médico Cirujano, Universidad de la Salud del Estado de Puebla (USEP), Puebla 72000, Mexico

**Keywords:** breast cancer, epi-drugs, epigenetic editing, epigenome

## Abstract

Epigenetic modifications are known to regulate cell phenotype during cancer progression, including breast cancer. Unlike genetic alterations, changes in the epigenome are reversible, thus potentially reversed by epi-drugs. Breast cancer, the most common cause of cancer death worldwide in women, encompasses multiple histopathological and molecular subtypes. Several lines of evidence demonstrated distortion of the epigenetic landscape in breast cancer. Interestingly, mammary cells isolated from breast cancer patients and cultured ex vivo maintained the tumorigenic phenotype and exhibited aberrant epigenetic modifications. Recent studies indicated that the therapeutic efficiency for breast cancer regimens has increased over time, resulting in reduced mortality. Future medical treatment for breast cancer patients, however, will likely depend upon a better understanding of epigenetic modifications. The present review aims to outline different epigenetic mechanisms including DNA methylation, histone modifications, and ncRNAs with their impact on breast cancer, as well as to discuss studies highlighting the central role of epigenetic mechanisms in breast cancer pathogenesis. We propose new research areas that may facilitate locus-specific epigenome editing as breast cancer therapeutics.

## 1. Breast Cancer Pathogenesis and Subtypes (Histological and Molecular)

Breast cancer (BC) is the most prevalent cancer in women worldwide. Over the years, the global cancer landscape has changed, and the incidence of BC has been rising. According to the global cancer statistics 2020 from the international agency for research on cancer (IARC), and the American Cancer Society (ACS), breast cancer has overtaken lung cancer worldwide as the most diagnosed cancer, with an estimated 2.3 million new cases (11.7%), followed by lung (11.4%), colorectal (10.0%), prostate (7.3%), and stomach (5.6%) cancers [1]. BC is a multifactorial and complex disease that arises from the interplay between genetic factors and epigenetic dysregulation of critical genes modulating important cellular pathways [2]. BC can occur in any cell of the mammary glands with a wide range of histopathological and morphological subtypes, which often determine the BC treatment strategies. Generally, BC is either a non-invasive, which is confined to the epithelial components of the breast (carcinoma in situ), or invasive carcinoma. The invasive or infiltrative carcinoma is often caused by the abnormal proliferation of neoplastic cells in breast tissue, leading to the penetration of tumor cells from the duct wall into the stroma [3]. Both invasive and in situ carcinoma can be further classified as ductal and lobular, based on the site from which the tumor is originated. In ductal carcinoma in situ (DCIS), the epithelial cells lining the milk ducts become cancerous, but they do not spread into the surrounding breast tissue. On the other hand, when DCIS spread through the wall of the ducts into the nearby breast tissue it is known as invasive ductal carcinoma (IDC), which is the most common form of invasive breast cancer. Lobular carcinoma in situ (LCIS) is a rare type of breast cancer in which cancer cells form in the milk-producing glands (lobules), but they are confined within the breast lobules [4]. However, when cancer cells break out of the lobules to spread through the lymph nodes and other parts of the body, it is known as invasive or infiltrating lobular carcinoma (ILC) (Figure 1). In addition to the four main cancer subtypes mentioned above (DCIS, IDC, LCIS, ILC), there are two other main BC subtypes, including inflammatory breast cancer (IBC) and metastatic breast cancer (MBC). IBC is an aggressive and fast-growing BC subtype in which cancer cells mostly infiltrate the skin and lymph vessels of the breast. Meanwhile, in MBC, cancer cells have spread through other parts of the body such as the liver, lungs, bones, and brain. It is estimated that metastasis is responsible for almost 90% of cancer-related deaths [5,6,7], a hallmark ability of tumor cells to disseminate to distant organs throughout the body [8,9]. Several other studies have classified more rare subtypes of breast cancer with different behaviors and prognoses which help better define the characteristics and course of the disease [3,10,11]. It is worth mentioning that whether the BC is invasive or non-invasive will determine the patient’s treatment strategies and how they may respond to the treatments. Moreover, in a few cases, both the invasive and non-invasive BC can be detected in the same specimen, or a patient can be diagnosed with BC occurring in both breasts at the time of diagnosis. This form of BC is known as bilateral breast carcinoma (BBC), affecting 2–5% of all breast malignancies [12]. 

Despite serious efforts that have been made to assess the metastasis process in breast cancer in recent years, challenges remain for the diagnosis of breast cancer in the early-stages [13,14]. Although metastatic breast cancer (MBC) is considered incurable with currently available therapies, the discovery of new epigenetic biomarkers can lead to BC detection at early stages. Besides, drugs targeting specific epigenetic aberrations (epi-drugs) can open new horizons and provide promising tools for new-generation breast cancer treatment. 

On the molecular level, breast cancer is classified into 5 major subtypes, as defined by proliferation index (Ki-67) and hormone receptors (HR) expression, including estrogen receptor (ER), progesterone receptors (PR), and human epidermal growth factor receptor 2 (HER2). These 5 subtypes include luminal ER/PR positive (luminal A and luminal B), HER2 enriched, normal-like, and triple-negative receptors (basal-like) (Figure 1) [15]. Estrogen and progesterone are steroid hormones that have a significant role in regulating female reproductive and sex organ development. ER is one of the most common biomarkers used for BC prognosis. According to studies among all the BC subtypes, around 75–85% of them are ER-positive (ER+), which is less aggressive than ER-negative (ER−) BC [16]. Luminal A is a subtype of BC that is positive for hormone-receptor ER and/or PR, but negative for HER2 with the low level of Ki-67, whereas luminal B subtype is hormone receptor-positive (ER+ and/or PR+) and is either HER2 positive or HER2 negative, but with high levels of Ki-67. Luminal A subtype, however, has a better prognosis and a higher survival rate compared with the luminal B subtype [17,18,19]. HER2 or Human Epidermal Growth factor 2 is a transmembrane protein that functions as a tyrosine kinase receptor and is located in the epithelial cells of the mammary glands. HER2 enriched breast cancer subtype is hormone-receptor negative (ER−/PR−) and HER2 positive (HER2+) with high Ki67 index, incorporating around 20% of all the BC subtypes [20], and has been considered the subtype with the worst prognosis and lowest survival rate. However, with the new advances in clinical treatments and therapeutic approaches, it can be handled more successfully nowadays [21]. Normal-like subtype of breast cancer shares the same characteristics as those observed in the luminal A subtype, which is hormone receptor-positive (ER+/PR+), HER2 negative (HER2−), and low levels of Ki-67. Although both luminal A and normal-like subtypes have a good prognosis (higher survival rate), the prognosis for normal-like is, to some kind, worse than that observed in luminal A subtype (Figure 1) [22]. Triple-negative breast cancer, or TNBC, is the most aggressive and most heterogeneous subtype which targets women at younger ages and has a higher prevalence in women of African ethnicity [22]. This subtype of BC is negative for hormone-receptors (ER−/PR) and HER2 (HER2−) receptor and shows the poorest prognosis among all other breast cancer subtypes [17,22,23,24].

**Figure 1 ijms-23-09521-f001:**
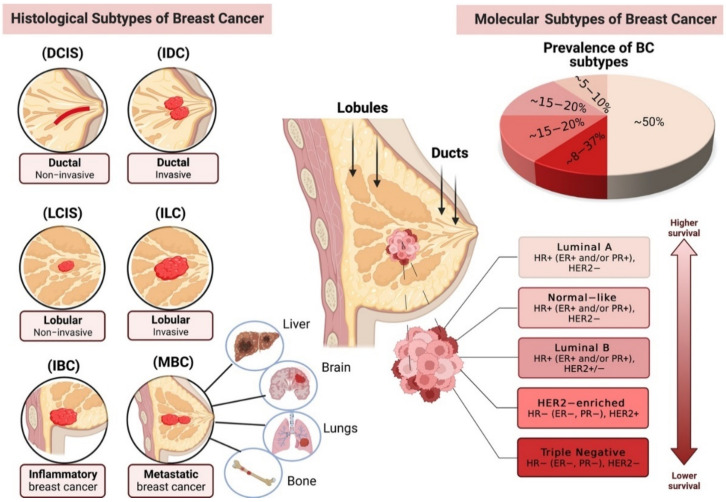
Main histopathological (**left**) and molecular (**right**) subtypes of breast cancer (BC). (DCIS—ductal carcinoma in situ, IDCS—invasive ductal carcinoma, LCIS—lobular carcinoma in situ, ILC—invasive lobular carcinoma, IBC—inflammatory breast cancer, MBC—metastatic breast cancer, HR—hormone receptors, ER—estrogen receptor, PG—progesterone receptor, HER2—human epidermal growth factor receptor) [3,17,22,25]. Figure created using BioRENDER.com. Accessed on 13 March 2022.

For a long time, breast cancer was regarded as a genetic disorder arising from mutations in key regulatory genes such as *BRCA1* and *BRCA2* [26]. However, discoveries in breast cancer in the last decades have shifted the focus away from genetics to epigenetics. For instance, recent studies have reported the role of aberrant DNA methylation in several cases of breast cancer [27,28], as well as other studies which have highlighted the epigenetic modifications as the major cause of breast cancer initiation and progression [29,30]. Breast cancer is often characterized by various hallmarks, including sustained proliferative signaling, activation of invasion and metastasis, evading immune destruction, apoptotic-resistance phenotype, genome instability and mutation, cellular energetics dysregulation, evading growth suppressors, induction of angiogenesis, vascular or lymphatic system invasion, and tumor-promoting inflammatory microenvironment [31]. Remarkably, the chronology and the contribution of cancer hallmarks differ according to cancer type, tumor composition, microenvironment, immune system interactions, and origin of cancer tissue. Some of the cancer hallmarks may have overlapping properties such as apoptosis resistance and avoiding growth suppressors, which result in sustained proliferative signaling of tumor cells. Other hallmarks might be required at distinct phases of tumor development. For example, metastasis activation and invasion occur later in tumorigenesis. Many of the mentioned hallmarks have tight links to epigenetic alterations affecting the expression of proto-oncogenes/tumor suppressors, which occur during the initiation and progression of breast cancer (Figure 2). Moreover, recent studies indicated that abnormal epigenetic modifications give rise to oncogenic properties on every cancer hallmark [32,33]. 

## 2. Epigenetic Modifications in Breast Cancer

Epigenetics is the study of heritable changes in gene expression that occur independently of alterations in DNA sequence. Epigenetic modifications affect molecular pathways during development, differentiation, and tumorigenesis, which are known to be triggered by age, diet, stress, hormones [38,39,40], as well as exposure to other environmental stimuli or factors known as exposome, as recently shown to significantly influence epigenetic modification [41]. Moreover, epigenetic alterations can be inherited through cell divisions and, in some cases, can be maintained over long periods of time or transmitted down through generations [42,43]. Epigenetic mechanisms can alter the gene activity at the transcriptional, post-transcriptional, and subsequent translational and post-translational levels, and have been linked to multiple diseases, including different types of cancer [44,45,46,47,48]. The main epigenetic modifications include DNA methylation, histone modifications causing chromatin remodeling, and non-coding RNAs (ncRNAs) regulating gene expression at post-translational levels. Unlike genetic changes, epigenetic changes are considered reversible, thus making them potentially promising therapeutic targets for disease treatment. Epigenetic therapies are considered to reactivate the expression of genes that have undergone epigenetic silencing, hence reprogramming the cancer cells. A better understanding of epigenetics in breast cancer may result in improved diagnosis, treatments, and prognosis. 

### 2.1. DNA Methylation in Breast Cancer

DNA methylation is catalyzed by a conserved set of enzymes known as DNA methyltransferases (DNMTs), which transfer a methyl group from S-adenosylmethionine (SAM) to the fifth carbon of cytosine residue to form 5-methylcytosine (5-mC) [49,50]. Methylation of DNA, mainly of CpG islands (CGIs) on the promoter region, is a common epigenetic mechanism that cells often use during differentiation, development, or disease initiation and progression to block transcription. The repressive effects of DNA methylation are mediated in large part by the methyl-CpG binding proteins (MCBPs). MCBPs specifically bind to CpG methylated DNA and are associated with the larger complexes containing histone deacetylases (HDACs) to “erase” the transcription-activating histone acetyl marks and hence, block transcription [51]. So far, numerous studies have demonstrated that hypermethylation of tumor suppressor genes [52], and/or hypomethylation of oncogenes [53,54] in tumors relative to non-tumorous tissues, is a common feature of a variety of cancers. Moreover, genome-wide methylation profiling studies have linked DNA methylation with the susceptibility to develop breast cancer [54,55], lower global DNA methylation in peripheral tissue being an indicator of an individual’s susceptibility to breast cancer [56,57]. Many of the detected hypermethylated genes are shown to act as tumor suppressors such as *P16* (cyclin-dependent kinase inhibitor 2A), *BRCA1* (breast cancer gene1), and *ATM* (ataxia telangiectasia mutated). In the last few years, there has been increasing interest in using blood samples to measure DNA methylation in cancer studies, with the most robust candidate genes *ATM* and *BRCA1* showing associations between breast cancer risk and methylation changes [58,59]. In addition, another study that conducted methylation microarray analyses from peripheral blood DNA across various genes, including *ATM*, *TP53*, *CDH1*, *BRCA1*, *BRCA2*, *MLH1*, and *CHEK2*, showed that hypermethylation of *ATM* was associated with an estimated threefold increased risk of breast cancer development [60]. Interestingly, a recent genome-wide methylation profiling study in both canine mammary tumor (CMT) and peripheral blood mononuclear cells (PBMCs) using reduced representation bisulfite sequencing (RRBS) led to the identification of 341 and 247 promoters of differentially methylated genes (DMGs) in CMT tissues and PBMCs, respectively [61]. These results indicated that genes related to apoptosis and ion transmembrane transport were hypermethylated, but cell proliferation and oncogene were hypomethylated in tumor tissues. In addition, their analysis further revealed significant changes in DNA methylation patterns of the subset of immune cells and host defense system-related genes in PBMCs, especially chemokine signaling pathway-related genes. Furthermore, several CMT tissue-enriched differentially methylated regions (DMRs) were identified from the promoter regions of various microRNAs (miRNAs), including *cfa-miR-96* and *cfa-miR-149*, which were reported as cancer-associated miRNAs in humans. Overall, these findings suggest that genome-wide hypomethylation [57] and gene-specific hypermethylation mainly of CGIs on promoters which typically act to repress gene repression, mainly of tumor suppressor genes [62,63], may appear at very early stages before diagnosis. Hence, they could be used as epigenetic biomarkers for early detection or risk of breast cancer development.

Since DNA methylation is catalyzed by the family of enzymes known as DNA methyltransferases (DNMT1, DNMT3A, DNMT3B, and DNMT3L), various studies have been conducted to understand aberrant DNA methylation patterns in association with the expression of DNA methyltransferases in different types of cancer [64,65,66,67]. In general, the impact of DNMT aberrations in the promotion of tumorigenesis is still controversial, and relevant targeted therapies for DNMTs are still under exploration [68,69]. Nevertheless, multiple studies revealed the increased expression of *DNMTs*, including *DNMT1*, *DNMT3a*, and *DNMT3b*, in breast cancer tissues, suggesting the involvement of the DNMTs during breast carcinogenesis [51,70]. Currently, two DNMT inhibitors (DNMTi) have been approved by the US Food and Drug Administration (FDA), including decitabine and azacytidine and their combinations with other anticancer agents, are being tested as therapeutic options for multiple solid cancers [71]. However, there is a crucial need to better optimize and improve the treatment strategies for DNMTi, as these inhibitors are not specific and are associated with the development of resistance, toxicity, severe side effects, and none or partial treatment responses. In another study, it was demonstrated that inhibition of DNMTs activity via the application of natural compounds such as EGCG and genistein could restore or reactivate the DNA hypermethylated tumor suppressor genes such as *CDKN1A* (*p21*). Hence, they strongly inhibit the formation of tumors in breast cancer [72]. In conclusion, DNA methylation is an important factor contributing to the pathogenesis of various types of cancer, including breast cancer. However, epigenetic therapy for targeting DNMTs to restore the tumor suppressor genes’ activity should be applied with caution in an isoform-specific and tissue-specific manner.

### 2.2. Histone Modifications in Breast Cancer

The modification of histone proteins is a crucial post-translational process that modulates the chromatin structure and plays a key role in gene expression. Histone modifications (HMs) are known to affect fundamental cellular processes, such as gene transcription, DNA replication, DNA recombination, and DNA repair [73]. Moreover, histone modifications are known to be connected to cancer initiation and progression [74,75,76]. Thus, a new terminology, ‘histone onco-modifications’, is proposed to describe post-translational modifications of histones, contributing to the cancer development [74]. Histones are chromatin proteins and basic components of nucleosomes in eukaryotic cell nuclei, which are essential for the packaging of the genomic DNA into compact structures. There are 5 types of histones, including H2A, H2B, H3, and H4, with their ability to heterodimerize in specific pairs, and H1 (linker) histone which links adjacent nucleosomes [77]. Numerous histone modifications are expected to exist, which can serve as markers of diagnosis in various cancer types [78,79]. So far, different types of histone modifications have been discovered and classified, including acetylation, methylation, phosphorylation, and ubiquitylation, which are the most well-understood, while O-GlcNAcylation, citrullination, crotonylation, ADP-ribosylation, sumoylation, deamination, formylation, propionylation, butyrylation, and proline isomerization are more recent discoveries that have yet to be thoroughly investigated [75,80,81,82,83,84]. Nevertheless, the two most studied histone modification mechanisms are namely methylation and acetylation. Each of these histone modifications is added or removed from histone amino acid residues by a specific set of enzymes.

Apart from DNA, which can be methylated to block transcription as described in the previous section, histone proteins can also be methylated, which can cause chromatin remodeling and affects gene expression. Histone methylation occurs mostly on arginine and lysine [85,86], and in rare cases on histidine residues of histone proteins [87,88]. Lysines are known to be mono-, di-, or tri-methylated [89,90]. While arginines are reported to be mono- or dimethylated [91], histidines, on the other hand, are found to be monomethylated [92]. Histone methylation can cause either transcriptional repression or activation. Generally, methylation of histones on H3K4, H3K36, and H3K79 is thought to mark active transcription, whereas H3K9, H3K27, and H4K20 methylation are associated with silenced chromatin states [93,94]. Intriguingly, it was shown that there is a crosstalk between histone and DNA methylation mark and that methylation of histones on H3K9 and DNA was shown to be strongly associated [95,96]. Therefore, the relevance of H3K9 methyltransferases dysregulation can expand to DNA methylation. In addition, histone methylation has been implicated in chromatin remodeling, DNA replication, and DNA damage response repair (DDR) [79,97]. Histone methylation is a dynamic and reversible process that is regulated by two groups of enzymes, including histone methyltransferases (HMTs) and histone demethylases (HDMs), which add and remove methyl groups on lysine and arginine residues within histone proteins, respectively. Recent studies indicated that HMTs might have an impact on tumor progression and metastasis. For example, it was shown that PRMT1 (a targeted HMT) could bind to the *ZEB1* promoter, which mediated histone methylation to promote the migratory and invasive behaviors in breast cancer cells [98]. In addition, PRMT1 could specifically increase the expression *ZEB1*, but not the rest of the crucial EMT markers such as *T**WIST1*, *SNAI1*, and *SNAI2*. Additionally, the study unraveled a dual role and a potential therapeutic value for PRMT1 in the modulation of both EMT and senescence via regulating *ZEB1*. In another study, the expression of several histone methylation markers, including lysin methylation (H3K4me2, H4K20me3), and arginine methylation (H4R3me2), was shown to be negatively correlated with tumor grade, as evaluated in 880 well-characterized BC patients. This was further validated using immunofluorescence staining and western blotting [99]. Besides, the high relative levels of global histone methylation were shown in this study to be associated with a favorable prognosis, which could be detected almost exclusively in luminal-like breast tumors (93%). Another study demonstrated that G9a, a histone methyltransferase responsible for histone H3 lysine 9 (H3K9) mono- and demethylation, is required for EMT-induced DNA methylation at the E-cadherin *(CDH1)* promoter in three model cell lines, and in Claudin-low breast cancer CLBC [100]. Thus, it results in decreased *CDH1* expression, a hallmark of EMT and a major contributing factor to early metastasis and poor patient survival associated with BC. Conversely, knockdown of G9a could restore *CDH1* expression by suppressing H3K9me2 and blocking DNA methylation which resulted in cell migration and invasion inhibition in vitro and suppression of tumor growth and lung colonization in models of CLBC metastasis in vivo. Despite extensive epigenetic research in breast cancer, studies on how histone methylation regulates BC progression and metastasis are still in infancy. Therefore, further investigations are required to unravel its precise underlying mechanism and contribution to breast cancer, which could define novel diagnostic and prognostic markers.

Histone acetylation is another well-known histone post-translational modification (PTM) that is found to have a wide role in the transcriptional regulation of genes [101,102]. This dynamic process is the outcome of the balance between histone acetyltransferases (HATs) known as “writers” and histone deacetylases (HDACs) or “erasers”. Histone acetyltransferases (HATs) install acetyl groups onto lysine residues of histone tails, which promotes open chromatin structure and leads to increased chromatin accessibility and hence increased gene expression. Conversely, histone deacetylases (HDACs) remove acetyl groups from histone tails, leading to a more compacted chromatin, lower transcriptional accessibility, and transcriptional repression [103,104]. Generally, histone acetylation is known to mediate both repression of tumor suppressors and activation of the proto-oncogene [101,102,105,106]. 

EP300 is one of the most studied HATs, encoded by the E1A binding protein P300 (*EP300*) gene which mediates histone and non-histone protein acetylation, and it is involved in gene activation [46,107]. Moreover, high expression and activity of EP300 was demonstrated to be associated with various diseases and malignancies, including pulmonary fibrosis [46], nasopharyngeal carcinoma (NPC) [108], hepatocellular carcinoma (HCC) [109], non-small cell lung cancers (NSCLC) [110], prostate cancer (PCa) [111], and breast cancer (BC) [112]. Moreover, EP300 is associated with several neurodegenerative disorders [113]. Mice expressing a truncated EP300 protein in the hippocampus, amygdala, cortex, and cerebellum were shown to have memory deficits in contextual fear conditioning and object recognition tasks [114]. Additionally, knockdown and pharmacological inhibition of EP300 were shown to reduce tumor growth and metastasis in vitro and in a xenograft mouse model of BC in vivo [112,115]. In addition to EP300, GCN5 is another family member of HATs that was found to play a key role in the TGF-β/SMAD signaling pathway in breast cancer cells [116]. Interestingly, knockdown of GCN5 was shown to inhibit EMT and decrease the migration and invasion of MDA-MB-231 breast cancer cells, which was associated with decreased expression of *p-STAT3*, *p-AKT*, *MMP9*, and *E2F1*, and increased expression of *P21*. Furthermore, PCAF, also a member of the HATs family, regulates EMT and promotes cancer metastasis via the BRD4-axis [117]. Hence, inhibition of PCAF displays the potential to inhibit breast cancer growth.

The role of HATs is opposed by HDACs. Similar to HATs, HDACs have key roles in various cellular functions, and their misregulation is linked to multiple types of cancer. Up to now, eighteen different HDACs have been recognized in humans and divided into four classes, based on phylogenetic and sequence homology to yeast proteins Hda1 and Rpd3 [103]. The class I HDACs (HDAC1, HDAC2, HDAC3, and HDAC8) contain a deacetylase domain and are primarily localized in the nucleus of cells. They have sequence similarity to Rpd3. The class II HDACs have sequence similarity to Hda1 protein and are commonly divided into two subclasses: subclass IIa consists of HDAC4, 5, 7, and 9, and subclass IIb includes HDAC6 and 10 based on their primary structure. Moreover, members of class II HDACs are considered to have tissue- and stage-specific expression, and are also capable of shuttling between cytoplasm and nucleus in response to various regulatory signals [118,119]. The Class III Sir2-like proteins (SIRT1, SIRT2, SIRT3, SIRT4, SIRT5, SIRT6, and SIRT7) are protein deacetylases dependent on nicotine adenine dinucleotide (NAD), known to regulate transcriptional repression, recombination, cell-division cycle, microtubule organization, and cellular responses to DNA-damaging agents [120]. The sole member of Class IV HDACs is HDAC11, which shares sequence similarity to both Rpd3 and Hda1 proteins and exhibits enzymatic activity to a certain extent [121].

HDACs are shown to regulate gene expression in multiple ways through targeted recruitment, protein-protein interactions, and post-translational modifications [122]. Moreover, HDACs catalyze the removal of acetyl groups not only from histone proteins, but also from various nonhistone proteins which can act as transcription factors, regulators of DNA repair, recombination, replication modifiers, chaperones, and viral proteins [122,123]. For example, HDAC3 was shown to catalyze the deacetylation of the Notch1 intracellular domain (NICD1), thereby promoting NICD1 protein stability, which regulates the progression of T-cell acute lymphoblastic leukemia (ALL) [124]. Moreover, Leus et al. showed that HDAC3 could inhibit NF-κB lysine acetylation, thereby causing a proinflammatory effect [125]. In another study, HDAC1 and HDAC2 were shown to deacetylate TP53 independently, thereby suppressing its ability to induce apoptosis and activate target genes, including *P21* [126]. Thus, regulation of key cellular processes such as cell survival, cell cycle progression, and differentiation are among the most important roles of HDACs. In addition, HDACs can associate with various corepressor complexes such as SMRT, N-COR, NuRD, and mSin3, which can further interact with other epigenetic modifiers to regulate gene expression, such as DNA methyltransferases (DNMTs), histone methyltransferases (HMTs), and methyl-CpG-binding proteins (MBDs) [103]. In general, the altered expression or activity of HDACs is linked with various human cancers and poor prognosis [4,127,128,129,130,131,132]. Consequently, HDACs have been among the most promising therapeutic targets for cancer treatment, including breast cancer, over the past two decades [133,134]. Recent studies show that class I HDACs are differentially expressed in breast cancer tissues, *HDAC2* and *HDAC3* being strongly expressed in the most aggressive tumor subtypes [135]. Moreover, the expression of *HDAC2* was significantly associated with an overexpression of *HER2* and the presence of nodal metastasis. In general, preclinical and clinical data show that HDAC inhibitors (HDACis) can evoke different antitumor mechanisms in distinct BC subtypes by targeting histone and several non-histone proteins. Hence, HDACis can sustain the cellular acetylation profile and reverse the function of proteins responsible for BC development [136]. Furthermore, it was demonstrated that HDAC11 promotes breast cancer growth and dissemination from lymph nodes (LNs) through activation of *PRM2* (pro-metastatic) and inhibition of *E2F7* and *E2F8* (cell cycle suppressors) [137]. Moreover, they observed decreased methylation of the *HDAC11* promoter, which was correlated with its increased messenger RNA (mRNA) expression in the mouse model of lymph node metastasis (AxLN). Altogether, these findings propose that HDAC11 may itself be epigenetically modified in the context of the LN microenvironment. Finally, HDAC11 suppression resulted in reduced lymph node growth and significantly reduced tumorigenesis. However, the knockdown of HDAC11 using short hairpins RNA (shRNAs), as well as global inhibition of HDACs using vorinostat and entinostat which are not HDAC11-specific, led to increased distant metastasis from LNs in vitro and in vivo, suggesting the involvement of other HDACs and taking caution with the single-agent use of HDAC inhibitors during BC treatment. Although the interest in histone deacetylase inhibitor-based therapies for cancer treatment is increasing in preclinical studies, in breast cancer, the efficacy of different HDACis, including trichostatin A (TSA), valproic acid (VPA), suberoylanilide hydroxamic acid (SAHA), and suberic bishydroxamate (SBHA), has been demonstrated. A better understanding of the molecular mechanisms underlying each individual HDAC members on tumor initiation and progression in breast carcinomas are required for the possible identification of new therapeutic strategies.

### 2.3. Non-Coding RNAs (ncRNAs) in Breast Cancer

Numerous studies have delineated the role of non-coding RNAs in diverse cellular processes, including proliferation, migration, invasion, apoptosis, and stemness. Moreover, it was shown that ncRNAs such as long non-coding RNAs (lncRNAs) or microRNAs (miRNAs) can be packed into extracellular vesicles (EVs) such as exosomes, and transported to cells locally or systemically [138]. 

In addition, ncRNAs were demonstrated to play key roles in transcription, post-transcriptional processing, and translation [139]. IncRNAs are one of the major subgroups of ncRNAs larger than 200 nucleotides that modulate gene expression, mostly through interaction with proteins and nucleic acids [140,141].

Recently, it was shown that a group of lncRNAs (*HOTAIR*, *linc-ROR*, and *BCAR4*) can regulate metastasis in breast cancer [142]. Among them, *HOTAIR* recruits the Polycomb Repressor Complex 2 (PRC2) complex to specific target genes, which leads to an altered histone H3 lysine 27 methylation, and subsequent epigenetic silencing of metastasis suppressors, which increases cancer metastasis and invasion [143]. However, lncRNAs can function specifically to either promote or inhibit cancer invasion and metastasis. For instance, *MALAT1*, *NKILA*, and *ANCR* are known to suppress those mechanisms in breast cancer [142]. 

miRNAs are single-stranded small non-coding RNAs that regulate gene expression in a wide range of biological processes through binding to the 3′untranslated region (3′UTR) of protein-coding genes, causing their translational repression [144,145,146,147,148]. Recent studies have shown that microRNAs can also bind to protein-coding exons and promote gene expression in mammalian cells [149,150].

MicroRNAs have been reported to play fundamental roles in modulating breast cancer progression and metastasis [151,152]. 

Reports indicate an association between dysregulation of miRNAs and several human diseases, including obesity [153], cardiovascular disease [154,155], and cancer [156]. Changes in miRNA expression, such as mutation, amplification, or deletion on miRNAs loci, have been reported to be linked to human cancers [157], highlighting the regulatory role of miRNAs on human diseases. Lethal-7 (*let-7*) is a member of tumor suppressor miRNAs highly expressed in epithelial tissues, which was shown to target oncogenes such as *MYC*, *RAS*, *HMGA2*, *BLIMP1*, and *LIN28* [158,159]. *LIN28* is a proto-oncogene shown to be highly expressed in various cancers, including breast cancer, which can drive or accelerate tumorigenesis via a *let-7* family of miRNAs dependent mechanism. [160]. Recently, an interesting mechanism was discovered for a member of the *let-7* family named *Mirlet7d.* Singh et al. showed that *Mirlet7d* is involved in epigenome regulation and genome organizations via binding to ncRNAs in the nucleus, forming *Mirlet7d*–ncRNA duplexes, which are further bound by other proteins such as CD1, which in turn target the RNA exosome complex and the PRC2 complex to the bidirectionally active loci [161]. Finally, this multicomponent RNA–protein complex was initially formed by *Mirlet7d*, termed MiCEE, which tethers the regulated genes to the perinucleolar region and hence is essential for the proper nucleolar organization. In addition, *miR-200* is another family of tumor suppressor miRNAs, since its suppression in epithelial to mesenchymal transition (EMT) has been reported to be associated with a considerably higher risk of breast cancer and invasiveness [162,163,164]. *MiR-21* is miRNA highly expressed in breast cancer, and its upregulation is associated with poor prognosis [165,166]. It has also been reported that *miR-21* targets *PTEN* [167], which consequently promotes MCF-7 breast cancer cell growth [168]. *MiR-10b* interacts with HOXD10 and Krüppel-like factor 4 (KLF4) [169,170], which has been reported to be an oncogenic miRNA in metastatic breast cancer [169]. *MiR-335* is a family of miRNAs which is recognized as metastasis inhibitor through its interaction with the transcription factor SOX4 and the extracellular matrix protein TNC, also reported to be silenced in the breast cancer [171,172]. Another miRNA that is involved in breast cancer regulation is *miR-155*, known to bind *BRCA1*, a breast cancer susceptibility gene [173,174]. Besides, *miR-155* negatively regulates *SOCS-1* and *FOXO3a*, which modulate breast cancer development [175]. Additionally, *miR34-a* modulates breast cancer, since its downregulation is reported to be associated with cancer development and progression through the upregulation of *SIRT1* and *BCL2* [176,177]. Downregulation of *miR-205* is often associated with breast tumor metastasis. Additionally, *miR-205* depletion results in the propagation of breast tumor cells and metastasis by increasing the levels of ERBB3, VEGFA, and ZEB1 proteins [178,179,180].

### 2.4. Non-Canonical Epigenetic Modification in Breast Cancer

In addition to the major canonical epigenetic modifications dysregulated in BC, including DNA methylation, histone modification, and noncoding RNAs, there is an increasing number of reports on epigenetic modifications on RNA, including N1-methyladenosine, 5-methylcytidine, inosine, 2′-O-ribosemethylation, pseudouridine, and N6-methyladenosine (m6A). These modifications strongly emerge as novel non-canonical epigenetic modifications associated with tumorigenesis [181,182]. Among them, m6A is considered the most prevalent, and conserved internal transcript modification, especially in eukaryotic cells, substantially impacts RNA metabolism and is involved in the pathogenesis of many diseases, including cancer [183,184]. Besides, m6A was mentioned to play a key role in pre-mRNA processing, alternative splicing, nuclear export, stability, and translation [185,186].

m6A modification is installed onto mRNA by the m6A methyltransferase complex known as “writers”, including WTAP, RBM15/15B, METTL3/14, ZC3H13, and VIRMA, and can be recognized by m6A-binding proteins known as “readers”, including eIF3, YTHDC1/2, YTHDF1/2/3, IGF2BP1/2/3, and HNRNP. In contrast, demethylases, or “erasers” such as FTO, ALKBH3, and ALKBH5, were shown to remove m6A modification from mRNA, which is a dynamical and reversible biological process. For instance, Jia et al. discovered that fat mass and obesity-associated protein (FTO) shows efficient oxidative demethylation activity of abundant N6-methyladenosine (m6A) residues in RNA, thus acting as m6A eraser partially localizing with nuclear speckles [187]. Moreover, they showed that siRNA-mediated knockdown of FTO led to the increased levels of m6A in mRNA, while overexpression of FTO resulted in a decreased level of m6A in human cells, suggesting m6A in nuclear RNA as a major physiological substrate of the obesity-associated FTO. Intriguingly, the heterogeneous nuclear ribonucleoprotein A2/B1 (HNRNPA2/B1) was demonstrated to be increased in endocrine-resistant LCC9 breast cancer cells and modulating the miRNA transcriptome upon its overexpression in MCF-7 cells [188]. Transient overexpression of *HNRNPA2/B1* was associated with alteration on miRNA expression (upregulation of 148 and 172 miRNAs and downregulation of 88 and 172 miRNAs, 48 h and 72 h after transfection, respectively). Moreover, in the same study, the overexpression of *HNRNPA2/B1* resulted in reduced MCF-7 sensitivity to 4-hydroxytamoxifen and fulvestrant, suggesting a role for HNRNPA2/B1 in endocrine resistance in BC cells. Another study discovered that HNRNPA2/B1 binds m6A-bearing RNAs as well as m6A marks in a subset of primary miRNA transcripts in vivo and in vitro. Furthermore, HNRNPA2/B1 interacts with the microRNA microprocessor complex protein, namely DGCR8, which promotes primary miRNA processing [189]. Altogether, these studies highlight HNRNPA2/B1 as a reader of the N6-methyladenosine (m6A) mark in primary-miRNAs (pri-miRNAs) and also as a modulator of primary microRNA processing and alternative splicing. Recently, the m6A methyltransferase *METTL14* was shown to be significantly upregulated in BC tissues compared with normal tissues [190]. In addition, METTL14 gain- and loss-of-function were shown to regulate m6A levels in both MCF-7 and MDA-MB-231 cells. *METTL14* overexpression revealed the enhanced migration and invasion capacities of BC cells via m6A modification and *has-miR-146a-5p* expression. Conversely, treatment with the m6A inhibitor suppressed the enhanced cell migration and invasion observed in *METTL14*-overexpressed BC cells.

Circular RNAs (circRNAs) are novel single-stranded non-coding RNAs in which the 5′ and 3′ termini are covalently linked by back-splicing of exons from a single pre-mRNA forming a closed continuous loop [191]. Initially, these untypical RNA molecules were considered aberrant splicing by-products [192,193,194]. However, new studies suggest that circRNAs are abundant, evolutionarily conserved, and expressed at specific developmental stages, in specific tissues or cell types, and even in a disease-specific manner [195,196,197,198,199]. Today, circRNAs emerged as a new research paradigm in RNA biology, with more than 32,000 different annotated circRNAs identified in the humans [200]. Increasing lines of evidence show that circRNAs regulate a large number of cellular processes by acting as a competing RNA molecule sponging miRNAs, transcriptional regulators, anchors for circRNA binding proteins (cRBPs), molecular scaffolds, and sources for translation of small proteins/peptides [201]. CircRNAs are generated by back-splicing which requires canonical splicing machinery, including splice signal sites and spliceosomes, and is further modulated by m6A in a METTL3/YTHDC1-dependent manner [202,203]. CircRNAs formed from exons are predominantly localized to the cytoplasm [196]. However, some exonic circRNAs are mentioned to be distributed in the nucleus [204] or in extracellular vesicles (EVs), and can be detected in circulation and urine [205]. Nevertheless, circRNAs packaging, delivery, and absorption still remain elusive. CircRNAs represent ideal candidates for application as non-invasive biomarkers due to their high stability and easy detection in body fluids. In addition, circRNAs can be detected by RNA-sequencing (RNA-seq) and/or microarray techniques with RT-qPCR or digital droplet PCR [206]. So far, few circRNAs have been reported to serve as miRNA sponges in BC. For instance, *circEPSTI1* was found as a prognostic marker and mediator of triple-negative breast cancer (TNBC) progression, which binds to *miR-4753* and *miR-6809* as a miRNA sponge to regulate *BCL11A* expression and affects TNBC proliferation and apoptosis [207]. In addition, silencing of *circEPSTI1* inhibits TNBC cell proliferation and induces apoptosis in three TNBC cell lines. In another study, Liu et al. identified *hsa_circ001783* as a highly expressed circRNA in both BC cells and tissues, which is significantly correlated with heavier tumor burden and poorer prognosis of patients with breast cancer [208]. Intriguingly, *hsa_circ001783* was shown to promote the progression of breast cancer cells via sponging *miR-200c-3p* and serves as a novel prognostic and therapeutic target for BC. Conversely, knockdown of this circRNA remarkably inhibited the proliferation and invasion of breast cancer cells.

Overall, emerging evidence indicates that circRNAs may potentially serve as a required novel biomarker and therapeutic target for cancer treatment. However, the function of the majority of circRNAs remains elusive. In addition, little is known about the role of circRNAs in breast cancer. Hence, further investigations are required to initially identify the subtype-specific circRNAs and to unravel their biological role in regulating tumorigenesis in BC. 

## 3. Future Perspectives and Novel Strategies for Breast Cancer

Currently, diagnosis of BC at an early stage remains one of the biggest challenges in oncology. In addition, a great proportion of breast cancer in low- and middle-income countries (LMICs) is mentioned to be diagnosed at an advanced stage, mostly due to the ill-prepared and fragile healthcare systems ranging from 30% to 50% in Latin America to 75% in Sub-Saharan Africa [209,210,211]. Furthermore, delays in the treatment of BC have been associated with more advanced disease stage cancer at diagnosis and poorer survival [212,213]. Although current methods for BC diagnosis are mostly based on mammography, magnetic resonance imaging (MRI), ultrasound, computerized tomography, positron emission tomography (PET), and biopsy, nevertheless, these techniques have certain limitations, such as being costly, inability in detecting small cancers, especially in women with dense breast tissues, time-consuming, and not being suitable for young women. Moreover, the sensitivity of mammography is mentioned to be related to the age, personal history, ethnicity, radiologist’s experience, and technique quality [214]. Therefore, to improve the current diagnostic options, developing a high-sensitive and rapid early-stage breast cancer diagnostic method is urgent. Recently, with advances in computational and analytical techniques, researchers have gradually shifted their attention to breast cancer early detection through the development of specific biomarkers [214,215,216]. Hence, the identification of novel diagnostic and prognostic biomarkers will pave the way for the early detection of BC and provide better opportunities for its prevention and treatment, which intends to result in a major shift in the reduction of mortality and morbidity of BC worldwide.

### 3.1. Novel Diagnostic and Prognostic Epigenetic Biomarkers for BC Detection

The importance of the identification of biomarkers lies not only in their prognostic value, which determines the future course of a disease, but also in the fact that they can predict the patient’s response to a selected therapy. Thus, diagnostic biomarkers are required to screen and also classify BC patients. On the other hand, prognostic biomarkers are needed to predict the patient’s survival [217,218]. Although earlier studies on biomarkers have been mainly focused on non-epigenetic molecular mechanisms, recent studies have also evaluated the potential of epigenetic marks in both solid and liquid biopsies from breast cancer patients [219]. Even in the last decade, some biomarkers have been integrated into clinical practice. As an example, Kim et al. established a two-gene panel for BC detection, namely *RARβ* and *RASSF1A*, which was shown to detect DCIS/IDC with significant sensitivity and specificity of 94.1% and 88.8%, respectively [220]. In another study, a seven-methylated-gene panel comprising *APC*, *BRCA1*, *CCND2*, *FOXA1*, *PSAT1*, *RASSF1A*, and *SCGB3A1* could detect BC in serum with high sensitivity, specificity, and an accuracy of 95.55% [221]. Moreover, a novel panel of three DNA methylation markers, including *TDR10*, *PRAC2*, and *TMEM132C*, was recently established as a high-potential diagnostic and prognostic marker, mainly because of their high expression in BC tissues, particularly in estrogen-receptor (ER)-positive patients [222]. Solid biopsies are considered the current standard of care in clinical cancer management, providing useful information about tumor histology, molecular biomarkers, histological subtyping, and treatment planning with optimal cost-effectiveness ratio [219]. Nevertheless, they inherently show several limitations, including high cost, limited accessibility of primary tumors and metastatic tissue, limitation of genetic and molecular information to the biopsy area, tedious and time-intensive sample processing, the high-pain risk for the patients, and the possibility for misleading interpretation due to the tumor heterogeneity. Liquid biopsy (LB), on the other hand, permits investigating the disease features in a more comprehensive manner by sampling biofluids such as circulating-tumor cells (CTCs) [223], circulating tumor DNA (ctDNA) [224], circulating tumor RNA (ctRNA) [225], and EVs obtained from blood or other body fluids [226,227], in a monitoring-compatible manner. Cells release DNA into the circulation known as cell-free DNAs (cfDNAs). cfDNAs are degraded DNA fragments derived from a combination of necrosis, apoptosis, and active secretion from both cancer cells and non-cancer cells released into the blood plasma carrying genome-wide DNA information [228]. ctDNA, on the other hand, is a small fraction of cfDNA which can be derived from primary tumors and metastases and even circulating tumor cells (CTC) and should not be confused with cfDNA. Nevertheless, ctDNA offers a non-invasive approach for initial diagnosis and longitudinal treatment response by capturing tumor heterogeneity and resistance patterns [229]. Furthermore, studies indicate that the use of ctDNA to predict the risk of postoperative recurrence of NSCLC is a highly valuable method, and it is even more reliable if combined with the dynamic changes of the cfDNA [230]. 

Elevated cfDNA levels are associated with an increased rate of cancer relapse in colorectal cancer patients [231]. Recently, Garcia et al. proposed cfDNAs in plasma as a predictive and prognostic marker in patients with metastatic breast cancer [232]. In addition, they showed that the amount of total cfDNA and the number of CTCs are predictors of overall survival (OS). However, total cfDNA levels are the sole predictor for progression-free survival (PFS) and disease response when comparing the response to non-response to treatment. Additionally, they suggested the analysis of CTCs and cfDNA is more informative than the combination of two conventional biomarkers (*CA15-3* and *AP*) for the prediction of OS. Another recent study assessed the prognostic and predictive potential of the blood circulating cell-free DNA (ccfDNA) in early and advanced breast cancer [233]. The study consisted of 3 patient groups and 1 healthy control, including 150 and 16 breast cancer patients under adjuvant and neoadjuvant therapy, respectively, 34 patients recognized with metastatic BC, and 35 healthy volunteers. Their findings showed that elevated levels of the blood circulating cell-free DNA (ccfDNA) were statistically correlated to the incidence of death, shorter PFS, and non-response to pharmacotherapy in the metastatic BC group, but not in the other groups. Interestingly, they showed all three types of fragment sizes of ccfDNA derived from apoptosis (~160 bp), necrosis (larger than 10,000 bp), and active secretion from viable cells (2000 bp), as analyzed by size-profiling using capillary electrophoresis. Patients with increased tumor burden in the metastatic and neoadjuvant groups often show abundance in shorter fragments and a more fragmented pattern of distribution in relation to adjuvant and control groups. Furthermore, the same study evaluated the methylation status of 5 cancer-related genes, including *SOX17, WNT5A*, *KLK10*, *MSH2*, and *GATA3* in plasma ccfDNA of BC patients. The results show that methylation of *SOX17*, *WNT5A*, *KLK10*, or the simultaneous methylation of at least three genes, was detected more frequently in 3 patient groups than in the controls. In addition, the methylation of *WNT5A* was statistically significantly correlated to greater tumor size and poor prognosis characteristics in advanced stage disease with shorter OS. In the metastatic group, *SOX17* methylation was significantly correlated with the incidence of death, shorter PFS, and OS. They also showed that *MSH2* was more frequently methylated in the adjuvant and metastatic groups than in the controls, meanwhile, *GATA3* was more frequently methylated in the neoadjuvant group than in the control and adjuvant groups. Finally, they concluded that ccfDNA emerged as a highly potent predictive classifier in metastatic BC in combination with established clinicopathological features, which could help for early and accurate diagnosis and prognosis. Another recent study which implemented a large-scale prospective study of 259 participants reported a significant decrease in cfDNA blood level after (1) surgery, (2) surgery and radiation, (3) neo-adjuvant chemotherapy and surgery, and (4) at the end of all treatments [234]. However, cfDNA could not discriminate between benign and malignant BC after mammography. Finally, they concluded that reducing tumor burden by surgery and chemotherapy is associated with reducing cfDNA levels. However, in a minority of patients, an increase in post-treatment cfDNA blood level might be an indication of the presence of a residual tumor and higher risk. 

It is noteworthy to mention that a proportion of cfDNA originating from mitochondria termed mt-cfDNA is still not well characterized and can be present either in a naked form or associated with internal and external mitochondrial membrane fragments which provide a useful way to estimate tumor burden and BC progression [235]. Altogether, these studies indicate the immense potential of cfDNAs as a versatile biomarker for diagnostics, prognostics, and therapeutics in oncology. However, cfDNA has multiple limitations to overcome that restrict its possible application as a definite marker to detect BC. cfDNA is not specific and does not originate solely from tumor cells. It also comes from tumor microenvironment cells and other non-cancer cells from various parts of the body which can elevate in other malignancies and pathologies, such as inflammation, acute bacterial or viral infections, or severe chronic disease, tissue trauma, sepsis, and myocardial infarction [236,237,238]. Moreover, there are yet a number of unresolved questions that remain to be answered, including the nature of cfDNAs, their subtypes, and mechanisms of release, clearance in patients with cancer, as well as their association with the origin, aggressiveness, response to treatment, and metastatic tumor potential [239]. 

During the last few years, genome-wide association studies (GWAS) have detected thousands of single nucleotide polymorphisms (SNPs) associated with human complex diseases [240,241]. However, unraveling their molecular mechanisms and the biological functions underlying the pathogenesis of complex diseases remains a great challenge. Recently, one of the largest GWAS studies (NHGRI-EBI GWAS Catalog of published genome-wide association studies, targeted arrays) was detected over 7000 SNPs in BC [242]. Most of these identified SNPs are, however, localized within the non-coding genomic regions, and their physiological functions are yet to be determined. 

It is widely accepted that the genetic information which flows from the genome to the traits goes through different intermediate molecular layers, including genome, epigenome, transcriptome, proteome, and metabolome [243]. Thus, the novel strategies today for biomarker identification have evolved into a large-scale multi-omics data analysis, not only for the identification of key drivers of diseases but also to discover novel biomarker tools [244,245]. This type of analysis from solid and liquid biopsy samples can help to detect specific genetic mutations such as SNPs and differentially expressed genes (DEGs) combined with the identification of epigenetic aberrations, including DNA methylation, the potential main site of histone modifications, the histone tail acetylation patterns, and the circulating non-coding RNAs in specific tumor subtypes. Moreover, the integrative multi-omics approaches offer unparalleled opportunities to decode the underlying biology of complex diseases, including various types of cancer such as BC [244]. However, the identified specific epigenetic marks should be correlated to the BC subtype, disease stage, progression, and aggressiveness of the tumor to result in the development of a reliable biomarker with high sensitivity, specificity, and positive predictive value (PPV) [246]. In addition, extending the integrative multi-omics to draw a molecular signature of BC which is not solely based on conventional mRNA expression levels, but integrating mutation, copy number alterations (CNAs), epigenetic marks such as methylation, acetylation, mRNA, microRNA, proteomics, and metabolomics datasets can lead to the identification of novel BC subtypes which would be better translated to clinical tests and precision medicine to facilitate novel therapeutic opportunities. Furthermore, the heterogeneity of breast cancer is distinct between cancer subtypes. Thus, it is crucial to investigate the distinct epigenetic dysregulation in both cancerous as well as non-cancerous cells in the BC tumor microenvironment (TME) [247]. In addition, these studies can further help to unravel the underlying mechanisms of homeostasis of immune cells and the complex communication between tumor and microenvironment, which is not yet well understood. Once the comprehensive epigenetic modification profiles of the patients are identified, the epigenetic therapies using epi-drugs and/or epigenetic editing should be applied to reprogram or abrogate epigenetic aberrations associated with specific BC subtypes. 

### 3.2. Epigenetic-Modifying Drugs for Therapeutic Intervention in BC

Unlike genetic modifications which result in DNA sequence alterations, aberrant epigenetic modifications are potentially reversible. Hence, the application of epigenetic-modifying drugs to restore normal phenotype is feasible and provides excellent opportunities for therapeutic intervention [248,249]. So far, the application of inhibitors for histone deacetylases (HDACs) or DNA methyltransferases (DNMTs) in BC treatment have been assessed in numerous clinical trials to evaluate the efficacy of these drugs to overcome epigenetic alterations (https://clinicaltrials.gov). Nevertheless, azacitidine and decitabine (cytidine analogs) are the two DNMT inhibitors or hypomethylating agents which received approval by the US food and drug administration for the treatment of myelodysplastic syndrome (MDS), acute myeloid leukemia (AML), and chronic myeloid leukemia (CML), which were shown induce DNA demethylation [66,250,251,252]. Furthermore, their combinations with other anticancer drugs are being validated as therapeutic options for various solid cancers such as ovarian, colon, and lung cancer [71]. Intriguingly, estrogen receptor (*ER*) or progesterone receptor (*PR*) was shown to be epigenetically silenced by DNMT and HDAC in BC, which could be restored by the application of epi-drugs [253,254]. Recently, a comprehensive study on assessing the effects of decitabine in breast cancer revealed a range of responses to decitabine that could not be predicted based on mechanisms like demethylation of tumor suppressor genes and viral mimicry response [255]. Furthermore, they demonstrated that while decitabine induces genome-wide expression changes and demethylation using transcriptome and methylome analysis, these effects are not necessarily paired and do not correlate with sensitivity. Instead, they showed that the multiple gene expression changes induced by decitabine are either an indirect result of its demethylation or the effects of induced cell death, DNA damage, and immune responses. Thus, the antitumor effects of hypomethylating drugs encompass cytotoxic effects, apoptosis, growth arrest, differentiation, and inhibition of angiogenesis. Therefore, the efficacy of monotherapy with epi-drugs is not satisfactory, and combining epigenetic drugs with other therapies such as immunotherapy or chemotherapy for solid tumors seems to be a better option [256,257]. Similarly, HDAC inhibitors simultaneously induce acetylation of histones as well as non-histone proteins, involved in the regulation of gene expression in various cellular pathways, which thus can contribute to toxic and fatal side effects. Although various studies targeting HDACs in an isoform-specific manner delineated their valuable effects on correcting abnormalities in cell proliferation, migration, vascularization, and apoptosis in malignant cells, they could achieve enhanced clinical utility by reducing or eliminating the serious side effects associated with non-selective HDAC inhibitors [258,259].

### 3.3. Epigenetic Editing as a Valuable Tool for Rewriting or Erasing Aberrant Epigenetic Marks in BC

Epigenetic editing offers powerful tools for locus-specific erasing or rewriting an epigenetic modification aiming to reprogram or modulate its expression. In principle, epigenetic editing involves the fusion of epigenetic enzymes or their catalytic domains (CDs) (also known as epigenetic effector domain) with programmable DNA-binding platforms such as the clustered regularly interspaced short palindromic repeat (CRISPR) to target a specific endogenous locus (Figure 3) [260]. CRISPR-Cas9 was mostly used for conventional genome editing. However, it is no longer just a gene-editing tool. The application of CRISPR-Cas9 has exceeded in different areas, including gene regulation, epigenetic editing (when coupled to epigenetic modifiers), chromatin engineering, and imaging [261]. Nevertheless, safety and efficiency issues regarding double-strand breaks (DSBs) and off-target mutations are still crucial limitations regarding the clinical application of the CRISPR-Cas9 technology [262,263]. On top of that, prime editing has recently emerged as a new CRISPR-Cas9-based approach with high accuracy that has revolutionized the genome editing area [264,265,266]. In this approach, no DSB-dependent repair is required for accurate repairing of DNA, which prevails the limitations of conventional CRISPR-Cas9 genome editing tools [263,267]. The high efficiency of prime editing arises from the potentiality of targeting any part of DNA and its ability to induce insertion/deletion (INDELs) [263,268]. The prime editing method leans on reverse transcriptase (RT) and prime editing guide RNA (pegRNA), which are conjugated to the Cas9 nickase (Cas9n) to exert the required edits in the genome [263,266]. Even though prime editing is a newly evolved approach, reports show its increased application in a variety of studies [269,270,271,272,273,274,275]. Interestingly, it has recently been demonstrated that the application of the prime editing technology can reverse a TP53 missense C > T mutation (L194F) in the T47D luminal A breast cancer cell line [263], providing new hopes for breast cancer therapy and clinical treatment. Additionally, the application of CRISPR-Cas9 technology has further extended to the field of genome-wide screening which helps to identify gene alterations and their relation to cancer predispositions. In a recent study, using in vivo genome-wide CRISPR screening, Ji et al. demonstrated that *Lgals2* promotes TNBC via inducing the polarization and proliferation of M2-like macrophages through the CSF1/CSF1R cascade that consequently leads to the immune escape in TNBC [276]. In another study using genome-wide screening, TNBC susceptibility was unraveled as the result of interactions between oncogenic and tumor suppressor pathways. More specifically, it was discovered that the mTOR and Hippo signaling pathways are key regulators of TNBC [277]. Additionally, they demonstrated that synergistic mTOR/Hippo-targeted combination therapy apply torin1 and verteporfin inhibits the mTOR1/2 and YAP, respectively, which leads to a much more efficient anti-tumor activity compared with the application of the drugs individually. Mechanistically, torin1 stimulates micropinocytosis and an endocytic program, which results in the encompassing of extracellular fluid and catastrophic cell death, in turn promoting the verteporfin uptake and accelerating its pro-apoptotic impact on cancer cells [277]. As discussed in the previous section of this review, various genes are repressed by DNA methylation, which is associated with BC progressions such as *BRCA1* and *P16*. Targeted demethylation of BC-related genes in this case, or even deposit DNA methylation marks on BC proto-oncogenes at a specifically targeted locus via epigenetic editing tools, would offer unique therapeutic possibilities for breast cancer in the future. Moreover, transcription activator-like effector nucleases (TALEN, developed in 2010) is an efficient nuclease protein containing a TALE domain from Xanthomonas bacteria, and an endonuclease domain isolated from *Flavobacterium okeanokoites*, that can be engineered to target and modify specific sites of the genome in a living cell [278,279], hence making it a desirable tool for genome editing with high precision. TALEN appeared as the first genome editing tool to provide hope for human cancer treatment in 2015 [280]. Moreover, it has been the leading tool for the production of genome-engineered crops [281]. So far, TALEN has been used in multiple biological studies to produce targeted mutations across different species [282,283,284,285]. TALEN can be delivered as protein, as RNA, or as DNA, and can be transferred to a living cell or to an organism through different methods depending on the goal of the project and the recipient organism: (1) physical methods: microinjection and electroporation, (2) viral delivery, (3) bacterial delivery, and (4) chemical methods: liposomes or PEG [286]. For instance, there have been several reports on the effective transfection of TALEN DNA or RNA into mammalian cells via electroporation [287,288,289]. Other studies have shown the efficient application of viral vectors, such as lentivirus [290] and adenovirus [291,292], for the TALEN delivery method in mammalian cells. Bacterial-mediated approach is another efficient TALEN delivery method that has been reported in mammalian cells [293,294]. Alternatively, chemical approaches of TALEN delivery such as liposomes have been reported to be promising tools for genome editing [293,295,296].

### 3.4. Biorecognition Engineering and Partitioning of Anticancer Drugs to Improve BC Therapeutics

Biorecognition engineering is the process of creating bioreceptors or biosensors using various technologies that can interact with cancer biomarkers, and hence can be useful as diagnostic and therapeutic tools [297]. One type of biorecognition technology is antibody mimetic molecules such as aptamers and affibodies. Aptamers are oligonucleotides, either ssRNA or ssDNA, that bind to multiple targets with affinities and specificities comparable to antibodies, and can be modified to direct the release of supramolecular structures such as nanoparticles (NPs) [298]. For instance, EpApt-siEp is an aptamer against the Epithelial cell adhesion molecule (EpCAM), a marker for cancer stem cell (CSC), and a biomarker for metastatic BC [299,300]. Injection of MCF7 cells treated with EpApt-siEp into mice resulted in robust anti-proliferative activity and tumor regression without toxicity [300]. pRNA–HER2apt–siMED1 is an aptamer-based complex nanoparticle that binds HER2-overexpressing BC to silence the ER coactivator Mediator Subunit 1 (MED1) expression, a molecule implicated in tamoxifen resistance [301]. The HER2 RNA aptamer reduced the growth and metastasis capacities and made BC cells more susceptible to tamoxifen treatment [301]. 

Affibodies are engineered scaffold proteins based on a three-helix bundle domain designed to bind with high affinity to desired targets and obtain affibody-drug conjugates [302]. Xia et al. designed Z-M ADCN, a conjugate nanoagent self-assembled into nanomicelles, resulting in improved pharmacokinetics and in vivo targeting performance due to longer retention time in blood and higher drug accumulation in the tumor [303]. In addition, Yamaguchi et al. evaluated the combination of Near-infrared photoimmunotherapy (NIR-PIT) with a HER2 Affibody-IR700Dye and a trastuzumab-IR700Dye conjugate in BC [304]. IR700Dye is a photosensitizer that, when irradiated with near-infrared light (690 nm), causes damage to the cell membrane and cell death without an effect on normal cells [305]. Both conjugates, which target different epitopes on HER2 protein, enhanced the effect of NIR-PIT against HER2-positive BC cells, including those with low *HER2* expression, thus proposing a new therapeutic strategy for HER2^+^-patients resistant to trastuzumab [304]. 

Delving into the mechanisms that produce drug resistance is essential for understanding the activity and efficacy of the drug. Resistance to chemotherapy is a multifactorial problem that may be associated with mutations and overexpression of key proteins such as ER and MED1, respectively, in BC [306,307,308]. In addition to cellular mechanisms, there is currently a need to study the physicochemical properties of anticancer molecules as a new approach for improving drug design [309]. Intriguingly, Klein et al. demonstrated that antineoplastic drugs can be selectively partitioned and concentrated within phase-separated biomolecular condensates such that their pharmacodynamic characteristics are disturbed [310]. Biomolecular condensates are membraneless intracellular subcompartments that concentrate molecules. They can be formed through stoichiometric molecule-to-molecule binding, such as with chromatin scaffolding and histone modifications, or through liquid–liquid phase separation (LLPS) thermodynamically driven [311,312], regulate cell functions through various mechanisms, thus endowing cancer cells with traits [313]. BC cells that overexpress *MED1* are resistant to tamoxifen, due to an increase in the volume of transcriptional condensates where MED1 is incorporated, resulting in dilution of tamoxifen, which in turn affects its ability to dissociate the ERα-MED1 condensate, promoting oncogene transcription [310]. The study of the physicochemical properties of the compounds that allow them to concentrate in certain condensates not only leads to the design of new ways of modifying the drug but also improves the efficacy of the delivery systems or conjugates that direct it to specific condensates to overcome therapeutic limitations, such as drug resistance.

### 3.5. Nanoparticle-Based Drug Delivery to Overcome Drug Resistance in BC Treatment

Nano-drug delivery systems (NDDS) and nanomedicine have opened a new therapeutic era with specific advantages, including improved stability and biocompatibility, enhanced permeability and retention effect, and precise targeting over conventional drug delivery systems, hence reducing the adverse effects of anticancer drugs [314,315]. Moreover, Nanocarriers (NCs) or Nanoparticles (NPs) raise big hopes to overcome cancer-related drug resistance and are considered to enhance immunotherapy, as well as reverse the tumor immunosuppressive microenvironment [316,317,318]. In addition, NPs not only increase the half-life of drugs and enhance their accumulation into tumor tissues due to deep tissue penetration abilities [319], but also offer a platform for poorly soluble drugs to be encapsulated and delivered more efficiently into the circulation [320]. Nevertheless, the NPs are required to hold certain characteristics, such as the ability to escape the mononuclear phagocyte system (MPS), and the reticuloendothelial system (RES) clearance, penetration in extracellular matrix (ECM), remain stable in the vascular system until they reach the target, high-pressure penetration into the tumor fluid, and accumulate in the tumor microenvironment (TME) via tumor vasculature [321]. However, NPs should still pass through numerous physiological and biological barriers to extravasate into the tumor tissue and reach cellular and subcellular levels such as complex systems of several layers (epithelium, endothelium, and cellular membranes), ability to escape from the endo-lysosomal system, and enzymatic components [316]. Nevertheless, there are two defined targeting strategies for NPs drug delivery systems: passive targeting and active targeting [322,323]. Regarding active targeting, cancer cells are targeted specifically by means of direct interactions between ligands on the surface of NPs which bind to the overexpressed cluster of receptors, such as epidermal growth factor receptor (EGFR), on the surface of cancer cells [324,325,326]. The ligand–receptor interaction system allows NPs to distinguish targeted cells from non-targeted healthy cells, and therefore induces receptor-mediated endocytosis to successfully release therapeutic drugs inside targeted cells [327]. However, actively targeted NPs should reach their target in the first place to take advantage of this increased affinity and avidity, which remains a formidable challenge [328,329]. In addition, the active targeting mechanism relies on the overexpression of specific markers in tumor cells relative to the non-targeted healthy cells. However, healthy cells in certain tissues of the body may present the same amount of those specific markers, or even to a greater extent than in targeted-tumor cells, which leaves those non-targeted cells more vulnerable to drug toxicity [330,331]. Hence, efficient passive targeting seems to be a prerequisite for NPs designed to target tumor cells and seems less complex than the active targeting [328]. Passive targeting takes advantage of the enhanced permeability and retention (EPR) phenomenon and unique characteristics of solid tumors, such as leaky vasculature and impaired lymphatic drainage. Under certain circumstances such as hypoxia, inflammation, or neovascularization, the endothelium layer of the blood vessels becomes more permeable, which provides very little resistance to extravasation, and hence permits NPs and even macromolecules to leak or diffuse from such blood vessels and ultimately collect within cancer cells [316,332,333]. This process denotes the enhanced permeability part of the so-called enhanced permeability and retention (EPR) effect. However, when a tumor develops, the lymphatic function gets impaired, which leads to minimal interstitial fluid uptake and can cause the NPs retention as they are not cleared and stored in the tumor interstitium, which is the main cause for enhanced retention part of the EPR effect [334]. Hence, encapsulating these small molecules in nanosized drug carriers should be routinely carried out to improve tumor selectivity, entry into tumor cells, and reduce their side effects [335]. The size of NPs is an important factor in determining their characteristics and applications, which influences their extravasation and accumulation [314]. NPs that are larger than 100 nm are more likely to be cleared from circulation by phagocytes [336]. On the other hand, smaller NPs that are less than 10 nm in diameter can be easily filtered by kidneys, and tiny NPs less than 1–2 nm can leak rapidly from the normal vasculature to the damaged normal cells [337]. Thus, a size range of 10 to 100 nm in diameter is generally considered appropriate for cancer therapy. Another factor that affects the characteristics of NPs and influences their bioavailability and half-life is the surface material. For example, coating the surface of NPs with hydrophilic materials such as polyethylene glycol (PEG) was shown to reduce the opsonization and hence avoid their clearance by the immune system [338]. Moreover, hydrophilic NPs were shown to increase the drug circulation time with improved penetration and accumulation in tumors [338,339]. Moreover, the elasticity of NPs was shown to have potential benefits [340] with prolonged blood circulation for softer nanoparticles (10 kPa) compared to harder nanoparticles (3000 kPa) when applied in vivo. Furthermore, softer nanoparticles exhibit significantly reduced cellular uptake in vitro in immune cells (J774 macrophages), endothelial cells (bEnd.3), and cancer cells (4T1) [340]. In the meantime, regardless of NPs’ active or passive targeting strategies, their application in the clinical treatment of various cancer types remains unsatisfactory, with significant challenges such as complexity and sophistication in design, and lack of diagnostic imaging technologies to evaluate the targeting efficiency of NPs. Above all, the NP-mediated toxicity in the host due to inadvertent immune system recognition of nanoparticles can trigger a multi-level immune response, resulting in invasiveness of tumor cells and metastasis to distal organs. Hence, further investigations are required for a better understanding of the TME and the crosstalk between NPs and tumor immunity, identification of real biomarkers coupled with the development of better and more predictive pre-clinical animal models would help for more precise drug design and exploitation.

## Figures and Tables

**Figure 2 ijms-23-09521-f002:**
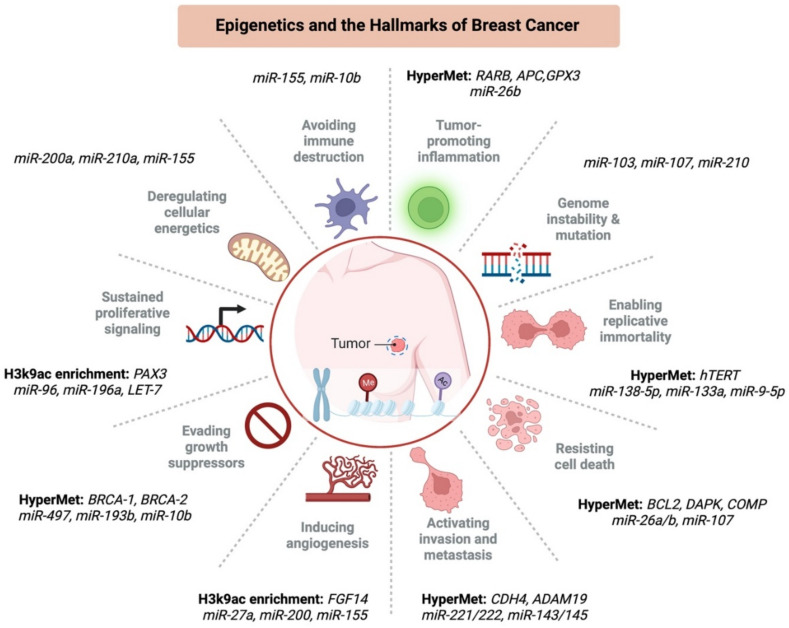
A collection of breast cancer hallmarks that characterizes the breast cancer phenotype with the central role for epigenetic modifications shaping the tumor outcome. Disruption of epigenetic processes modulates signaling pathways promoting breast tumorigenesis. Epigenetic alterations which are attributed to specific breast cancer hallmarks are depicted. (HyperMet; promoter hypermethylation) [6,34,35,36,37]. Figure created using BioRENDER.com. Accessed on 25 July 2022.

**Figure 3 ijms-23-09521-f003:**
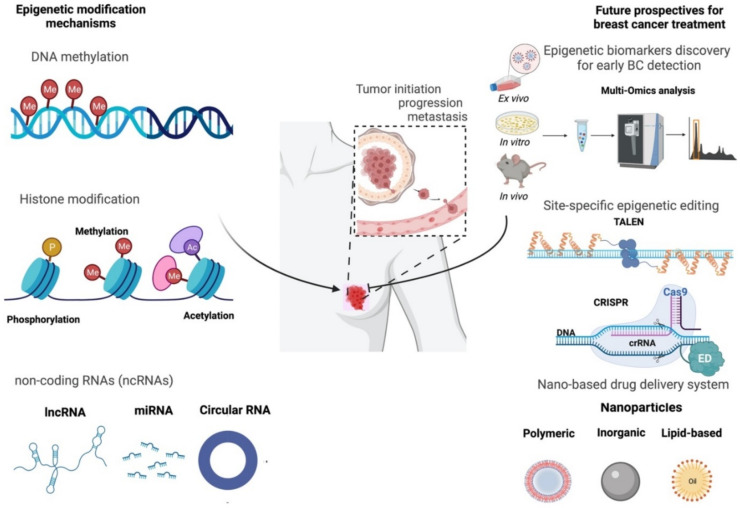
Epigenetic modifications detection and novel strategies for epigenetic editing in the therapeutic approaches for breast cancer. The aberrant changes of the epigenetic mechanisms, such as DNA methylation, histone modifications, and ncRNAs led to the activation of oncogenes and/or silencing of tumor suppressor genes affecting signaling pathways pivotal for cell maintenance, repair, and homeostasis. These epigenetic modifications regulate and contribute to every single cancer hallmark discussed (Figure 2), and eventually trigger breast cancer initiation, progression, and metastasis. The specific type of epigenetic modifications can be detected using solid and liquid biopsy sampling and the application of integrative analysis of multi-omics (genomics, proteomics, transcriptomics, epigenomics, metabolomics) approaches. The multi-omics data derived from human and mouse models of BC can help for the identification of aberrant epigenetic profiles, providing insights into cancer biology and rendering novel tumor epigenetic diagnostic, prognostic biomarkers, and epigenetic therapeutic tools with an important clinical value. Strategies for epigenetic corrections in BC patients are depicted. In site-specific epigenetic editing, CRISPR-Cas9 fused to the effector domain (ED) for epigenetic corrections, and TALEN strategies integrated with cell-specific promoters can be used to restore normal chromatin structure and correct specific epigenetic aberration localized to specific cell types. In addition, the application of a nano-based drug delivery systems (NDDs) increases the uptake rate of target cells or tissues and reduces enzyme degradation, hence improving the safety and effectiveness of drugs. Me—methylated DNA or methylated histone, P—phosphorylated histone, Ac—acetylated histone, ncRNAs—noncoding RNAs, miRNA—micro RNAs, BC—breast cancer, ED—effector domain. Figure created using BioRENDER.com. Accessed on 17 April 2022.

## Data Availability

Not applicable.

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
