# Peer review of "Advances of Epigenetic Biomarkers and Epigenome Editing for Early Diagnosis in Breast Cancer"

_ijms, 2022, doi:10.3390/ijms23179521_

Round 1
Reviewer 1 Report
Sarvari and co-authors provide a comprehensive review of common epigenetic mechanisms underlying breast cancer and therapeutic strategies to exploit these mechanisms. Their discussion of current epi-drugs and ongoing studies to implement new technologies is robust and timely. Figures clearly convey key points of the review article. Figure 3 includes circular RNA, which is not discussed in the ncRNA section. Please eliminate from the figure or provide discussion on circRNA.
Author Response
Many thanks for the reviewer's comments, please see the attachment.

Reviewer 2 Report
In this manuscript, Sarvari et al., reported a review about the recent advances of epigenetic biomarkers and epigenome editing for early diagnosis in breast cancer. Overall, the authors did an excellent job describing the current states about breast cancer epigenomics. The description of the current state regarding breast cancer is adequate, and the references are up-to-date.
I have a few suggestions:
1. The title emphasizes on “early diagnosis”, while in the main text the authors rarely mentioned “early diagnosis”. The authors need to discuss the application of using cfDNA as prognostic markers. For instance:
a) Circulating cell-free DNA in breast cancer: size profiling, levels, and methylation patterns lead to prognostic and predictive classifiers
https://www.nature.com/articles/s41388-018-0660-y
b) Cell-free DNA concentration in patients with clinical or mammographic suspicion of breast cancer
https://www.nature.com/articles/s41598-020-71357-4
2. In this manuscript, the authors mainly focused on the impact and mechanisms of DNA methylation, histone modifications, and ncRNA. However, the involvement of RNA modifications (mainly A6) has been reported in many cancer studies, including breast cancer. For instance,
a) METTL14 promotes the migration and invasion of breast cancer cells by modulating N6‑methyladenosine and hsa‑miR‑146a‑5p expression
https://pubmed.ncbi.nlm.nih.gov/32323801/
b) HNRNPA2/B1 is upregulated in endocrine-resistant LCC9 breast cancer cells and alters the miRNA transcriptome when overexpressed in MCF-7 cells
https://www.nature.com/articles/s41598-019-45636-8
3. For the “Epigenetic editing as a valuable tool for rewriting or erasing aberrant epigenetic marks in BC” section, “prime editing” (https://www.ncbi.nlm.nih.gov/pmc/articles/PMC9139850/) needs to be included.
4. Genome-wide CRISPR screen needs to be included, for example:
a) In vivo multidimensional CRISPR screens identify Lgals2 as an immunotherapy target in triple-negative breast cancer
https://www.science.org/doi/10.1126/sciadv.abl8247
b) In vivo genome-wide CRISPR screen reveals breast cancer vulnerabilities and synergistic mTOR/Hippo targeted combination therapy
https://www.nature.com/articles/s41467-021-23316-4
Author Response

(The authors gave the same response as above.)
